# Investment Decision of Blockchain-Based Traceability Service Input for a Competitive Agri-Food Supply Chain

**DOI:** 10.3390/foods11192981

**Published:** 2022-09-23

**Authors:** Pan Liu, Ziran Zhang, Ye Li

**Affiliations:** College of Information and Management Science, Henan Agricultural University, Zhengzhou 450046, China

**Keywords:** high-quality agri-food supply chain, blockchain-based traceability service, perceived quality and safety, investment decision

## Abstract

Competitive agri-food supply chain (hereafter, AFSC) is an important component of AFSC. In a competitive environment, more and more AFSCs use blockchain-based traceability services (hereafter, BBTS) to improve the traceability level of agricultural products. The investment rules concerning BBTS and the coordination rules in an AFSC are vital issues for many firms who want to adopt BBTS. To explore these laws, we constructed two competitive AFSCs, each of which included a supplier and a retailer. Considering the new changes in consumers’ perception of product quality and safety after using the BBTS, the demand function was modified. Then we built the income functions of chain members under three situations of investment. The research found that: (1) the improvement of customers’ perceived quality by using the BBTS can increase their benefits; (2) when decision-makers want to invest in the BBTS, they should pay attention to consumers’ perceived quality safety factor for their competitive products; (3) when the investment cost is greater than its threshold value, two competitive AFSCs should invest in the BBTS together.

## 1. Introduction

With the dramatic development of China’s economy and the improvement of people’s living standards, consumers’ demands for agricultural products are developing in a greener and safer direction. However, several produce scandals have occurred in the past decades, such as horse meat being found in several ground beef products in Europe in 2013, and 100,000-tons of expired meat being confiscated by Chinese authorities in 2015. In 2018, 44% of Canadian seafood products were mislabeled [1]. These quality and safety issues concerning agricultural products have seriously dampened consumers’ confidence in the quality and safety of agricultural products [2]. At the same time, it also stimulates consumers’ demand for quality traceability of agricultural products from farm to table [3,4].

To reduce these harmful food safety issues, the European Union, the United States, Australia, Canada, Japan, China, and other countries have issued laws and regulations to require the agri-food industry to build traceability service systems [5]. However, for traditional traceability systems, the information is not safe in the storage process. Because the data managers are also members of supply chains, when adverse data are found, they may modify the data. So, the credibility of the traditional traceability service is not high. As a distributed data structure, blockchain technology can share data on peer-to-peer networks [6,7]. In the blockchain environment, data were converted into digital code and stored in a shared database, with higher transparency and limited risk of deletion and modification. Blockchains cannot be usurped [8]. Blockchain technology has received extensive attention in the agricultural product traceability area [9].

In fact, some supply chain enterprises have adopted blockchain technology. For example, Wal-Mart put the blockchain-based traceability service (hereafter, BBTS) into the mango supply chain, and it took only 2.2 s to track the mangoes [10]. JD, Procter & Gamble, Amazon, and other companies also use blockchain. According to the measurement of consumption data from the JD Zhizhen chain, we found that applications of the BBTS could effectively improve repurchase rates and sales volumes. Therefore, more and more supply chain enterprises want to use the BBTS, but this also means an increase in expenditure. Thus, BBTS investment rules and laws have gained more attention from supply chain members. The competitive aspect of the high-quality agri-food supply chain (hereafter, AFSC) has, particularly, recognized the importance of the investment laws in using BBTS.

We investigate two competitive AFSCs’ investment strategies regarding BBTS and pay more attention to the following problems:How does the investment of the BBTS affect the profits and pricing of AFSC?What is the equilibrium strategy of supply chains where investment of the BBTS is concerned?When a supply chain invests in the BBTS, does it result in a competitive supply chain?

The research for this study mainly involved the following three aspects: first, operational management of competitive AFSCs; second, application of blockchain traceability service in AFSCs; third, AFSCs’ investment decisions based on the BBTS.

### 1.1. Operational Management of Competitive AFSCs

This research investigates the investment decision of competitive AFSCs. Dai et al. [11] studied the coordination and pricing of a competitive supply chain. They divided the competition of the supply chain into three models: upstream competition, downstream competition, and a mixed supply chain model with both upstream and downstream competition. They then designed a mixed competition mode to study the effect of traceability and product recall in the supply chain. Based on this mode, Dai believed that traceability investment is always beneficial to manufacturers, and the impact on retailers depends on the cost of traceability. Niu et al. [12] designed a co-appetitive supply chain model composed of a multinational firm located in a high tax region and an e-tailer that purchased and resold the multinational firm’s products. Choi et al. [13] designed a game mode of duopoly competition, and studied the effect of product information disclosure, based on blockchain technology, on the supply chain of rental service platforms. This study indicated that risk attitude is an important factor to accurately describe the impacts brought by blockchain technology.

In the field of AFSC, Ganeshkumar et al. [14] through a critical review of the literature in the field of AFSC management, divided it into three categories: policies affecting the segments of AFSC management, individual segments of AFSC and performance of supply chain segments. Joshi et al. [15] collected data from 1100 supply chain entities. It was found that low adoption capabilities and lack of uniform sustainable agri-business policy were the major factors influencing adoption of sustainable agri-business. Wang and Chen [16] studied the decision-making problem of portfolio contracts in the fresh supply chain and designed a model for concluding contracts for suppliers and retailers. They found that option prices have diverse impressions on supply chain members’ income. Wang and Zhao [17] also studied the investment decision of cold chain investing in fresh supply chains. Their results suggested that cooperative cold chain investment and collaborative pricing are the dominant strategies of the supply chain.

The above research includes the operation and management of competitive supply chains and agricultural product supply chains, but there are few studies on the operation and management of agricultural product supply chains considering market competition.

### 1.2. Application of Blockchain Traceability Service in AFSCs

This paper’s research involves the application of blockchain traceability service in supply chains. In recent years, with the promotion of national policy and consumers, the notion of a traceability service has gradually become an important part of AFSC management [18]. Rejeb et al. [19] analyzed big data in the context of AFSCs. Their findings indicated that traceability can improve food safety and bring sustainable AFSC benefits. Badia-Melis et al. [20] summarized the technological development of traditional traceability services, such as RFID, NFC, isotope analysis and DNA barcode, but did not show the development of blockchain technology. The above traceability services have deficiencies in traceability information security, transparency, and credibility.

Due to its traceability, digitalization and security, many scholars have studied how supply chains use blockchain technology [8,21,22,23,24,25,26]. Alkhader et al. [27] studies the adoption of blockchain to improve the traceability of products produced using additive manufacturing, guaranteeing the credibility of the source of transaction data and ensuring stakeholders’ trust. Yang et al. [28] used blockchain technology to design the supply chain traceability system for fresh agricultural products, which improved the transparency and credibility of supply chain information and made up for the shortcomings of traditional traceability services in transparency and credibility. Sun and Wang [29] studied the purchasing decisions of supply chain buyers considering traceability and found that buyers were more likely to purchase from suppliers with high traceability. Collart and Canales [30] believed that future research should focus on traceable economic sustainability data and the economic feasibility of blockchain technology, based on a case study of high-quality AFSC. Casino et al. (2020) [31] designed a traceability system of agricultural products based on blockchain, and evaluated the feasibility of the model through a specific study of a real dairy enterprise. Griffin et al. [32] used distributed ledger technology to detect cotton quality, track cotton data and coordinate supply chain management. Salah et al. [33] designed a traceability system for soybeans using blockchain and intelligent contracts. Through a survey of the wine supply chain, Saurabh and Dey [34] studied the factors that decision-makers were most concerned about in the adoption of blockchain technology.

At present, blockchain technology has been applied in business. The most famous case is the mango traceability system jointly developed by Walmart and IBM, which can greatly shorten the traceability time of mangoes. At the same time, Walmart also uses this technology for pork traceability [10]. IBM has also developed beef chains to track where beef comes from, and the technology is certified by the U.S. Department of Agriculture [35]. Based on blockchain technology, Carrefour has developed a traceability system for poultry agricultural products, like Walmart, which improves the traceability of chicken and eggs. Internet technology companies, such as Amazon and Oracle, have also developed blockchain traceability services [36]. Some Internet technology companies have developed blockchain traceability service platforms, and their agricultural product supply chain purchases blockchain traceability services from the platform. Ant Group developed Ant Chain, and Zhizhen Chain developed by JD Group can provide blockchain traceability services for agricultural product supply chains.

Through the above literature, it can be seen that many scholars have designed different blockchains for different AFSCs. To trace the origin of pork and mangoes, Walmart developed the pork chain and mango chain, respectively. The cost of developing different blockchain technologies for AFSCs is so huge that AFSCs cannot always afford it. The blockchain traceability service is based on the online blockchain platform, which can meet the traceability needs of various agricultural products. With the help of the platform’s blockchain traceability service, the cost of developing blockchains in the supply chain is reduced. Therefore, more and more agricultural product supply chains have adopted BBTS, such as Wuchang rice and West Lake Longjing tea. Therefore, building a BBTS platform suitable for a variety of agricultural products, and purchasing BBTS from the agricultural product supply chain, has become a future development trend.

### 1.3. AFSCs Investment Decision Based on the BBTS

Niknejad et al. [9] used bibliometric analysis to study the research and development of blockchain technology in agricultural supply chains in recent years and found that more and more scholars are interested in this field. The research shows that research on blockchains in the field of agricultural products is mainly divided into traceability system, blockchain technology and the benefits of blockchains.

Blockchain traceability system improves the traceability of agricultural products. The literature shows that traceability can increase the perception of product quality and enhance consumers’ confidence in agricultural products. Consumers have a higher willingness to pay for this kind of product [4,37,38]. Research shows that the main cost of supply chain investment in blockchain technology lies in variable costs, and the fixed cost of investment has little impact on the supply chain [39]. The research of Chen et al. [40] showed that consumers are sensitive to price and the selling prices of agricultural products are low. If the price of agricultural products significantly increases, due to the increase in the cost of anti-counterfeiting and traceability services, the sales volumes of agricultural products are reduced, which is not conducive to the profits of the supply chain. In the field of agricultural product supply chains, P. Liu et al. [41] considered freshness and greenness, and designed an agricultural product supply chain composed of a manufacturer and a retailer to study the integrated application of big data and blockchain. The research showed that when the investment cost was within a certain range, it was conducive to the profit of the supply chain. Stranieri et al. [42] studied the impact of blockchain technology on performance in agricultural supply chains, and argued that blockchain technology can bring profits and benefits to the supply chain, enhance quality attributes and improve supply chain management. Zhao et al. [43] believe that the application of blockchains in high-quality AFSC can improve traceability and quality safety. Wu et al. [44] studied the coordinated pricing problem of investing in blockchain technology in the fresh food supply chain, and designed a three-level supply chain model consisting of suppliers, third-party logistics and retailers. Their study found that whether to invest in blockchain Technology was related to consumer acceptance, cost sharing, and product deterioration.

Based on the analyses, the existing research has the following shortcomings: (1) the current research is mainly focused on blockchain technology, and there are few studies on the decision-making impact of blockchains in high-quality AFSCs; (2) at present, traceability research is primarily focused on the meaning of the traceability service and the technical level of the traceability system, but there are few studies on the economic profits of the traceability service and how competitive supply chain decisions about traceability service are made. (3) The above research does not involve the competitive decision-making of dual-channel AFSCs. In order to solve the deficiency of the above research, considering consumer price sensitivity and consumer perception of product quality and safety, this paper discusses the investment decision-making of the BBTS in competitive AFSCs under different investment situations.

The existing research mainly focuses on the technology and significance of blockchains, but there is little research on how the supply chain should invest in blockchain, especially in the current competitive business environment. The research on blockchain investment in competitive AFSCs is even less.

The aims of our study were to research the investment decision strategies of the BBTS in a competitive high-quality AFSC. Based on the above considerations, we firstly constructed two competitive AFSCs, each of them including a supplier and a retailer. Based on the new changes to consumers’ perceptions regarding product quality and safety after using the BBTS, the demand function model was modified. Then, considering BBTS costs and consumers’ perceptions on quality and safety, we built the income functions of chain members under three proposed situations.

There are three innovations in this paper: (1) a competitive demand function model was established based on consumers’ price sensitivity and consumers’ perception of product quality and safety; (2) we consider two competing supply chains, each supply chain has two investment options, and, thus, there are four situations. Due to the symmetry of two supply chains, we only need to study three situations. The N situation referred to a situation when the two supply chains did not invest in the BBTS, the I situation when only one supply chain invested in the BBTS, and the M situation when both the supply chains invested in the BBTS; (3) chain members’ profit models were established under the proposed three kinds of investment situations considering the BBTS costs and the change of consumers’ perception to product quality and safety.

The significance of this study is as follows: (1) considering changes in consumers’ perception to product quality and safety, we modified the demand function of agricultural products to enrich the research perspective of the demand function in AFSCs; (2) considering BBTS costs, we constructed chain members’ benefits functions under the three competitive supply chain models (N, I, M). This expands the theoretical research on investment decision-making of competitive AFSCs. Meanwhile, the results provide theoretical support for competitive AFSCs to invest in BBTS.

The article structure is shown in Figure 1.

## 2. Materials and Methods

### 2.1. Model Description

To study the BBTS investment strategy of competitive AFSCs, we constructed two competitive AFSCs, each of which include a supplier and a retailer, as shown in Figure 2.

Investing in the BBTS can improve consumers’ quality perception to fresh agricultural products. Investment situations of the BBTS are shown in Figure 3. The supply chain has two choices in decision-making, i.e., investing in the BBTS and not investing in the BBTS. We consider two competing supply chains, each supply chain has two investment options, so, thus, there are four situations as follows: (1) both supply chains do not invest in the BBTS; (2) supply chain 1 invests in the BBTS, supply chain 2 does not invest in the BBTS; (3) supply chain 1 does not invest in the BBTS, supply chain 2 invests in the BBTS; (4) both supply chains invest in the BBTS. Due to the symmetry of the two supply chains, we only need to study three situations: (1) neither of the two AFSCs invests in the BBTS, namely, the N situation; (2) only one AFSC invests in the BBTS, namely, the I situation; (3) both AFSCs invest in the BBTS, namely, the M situation.

The relevant parameters involved in the construction of the model and their descriptions are shown in Table 1.

### 2.2. Hypothesis

**Hypothesis** **1 (H1).**
*The quality of the two alternative agricultural products can meet the national quality and safety of agricultural products standards and the agricultural industry standards. Therefore, it is assumed that the qualities,*

$q_{i}$

*of the two agricultural products are the same.*


**Hypothesis** **2 (H2).**
*Consumers prefer to buy fresh agricultural products with trusted information because they think such products’ qualities are guaranteed. Blockchain-based traceability information is more reliable. Therefore, the supply chain investing in the blockchain technology can improve customers’ quality safety perception level to products. Therefore, we assumed*

$\lambda_{i}^{N}=\lambda_{2}^{I}<\lambda_{1}^{I}=\lambda_{i}^{M}$

*. The members of the supply chain make decisions with the goal of maximizing their own incomes.*


**Hypothesis** **3 (H3).***By**eliminating intermediaries, we can improve audit efficiency and reduce communication and financial costs. We assumed the optimization coefficient of cost is*θ.

**Hypothesis** **4 (H4).**
*The supplier has sufficient production capacity to ensure that the products are not out of stock. The members of the supply chain are risk-neutral and completely rational.*


**Hypothesis** **5 (H5).***Base on, we assumed that the demand of each supply chain is linear with its own retail price,**consumers’ perceived quality safety factor regarding the agricultural products, retail price and consumers’ perceived quality safety factor regarding competitive products. Therefore, the market demand of product 1 is*$D_{1}^{x}=a_{1}-p_{1}^{x}+\beta p_{2}^{x}+\lambda_{1}^{x}q_{1}-kq_{2}$*. The market demand of product 2 is*$D_{2}^{x}=a_{2}-p_{2}^{x}+\beta p_{1}^{x}+\lambda_{2}^{x}q_{2}-kq_{1}$*. Here,*$a_{1}>0$*,*$a_{2}>0$*. Therefore, we can get**the general formula of market demand, namely,*$D_{i}^{x}=a_{i}-p_{i}^{x}+\beta p_{j}^{x}+\lambda_{i}^{x}q_{i}-kq_{j}$. *Suppose*1>β>0*, demand is more sensitive to its own price.*$\lambda_{i}^{x}\geq 0$*,*k≥0*. The greater*β*and*κ*the stronger the competition between the two products. Suppose*$\lambda_{i}^{x}\geq k$*, the demand is more sensitive to its own quality level than to the substitute**quality level.*

**Hypothesis** **6 (H6).**
*In order to facilitate the research, it was presumed that the production costs of the suppliers in the two supply chains are zero. In order to enhance the competitiveness of the whole supply chain and obtain more revenues, both suppliers and retailers have an inherent motivation to invest in the BBTS. It was assumed that suppliers and retailers bear their own BBTS costs, and indirectly share the BBTS costs of each other by adjusting wholesale prices and ordering quantities.*


## 3. Results

### 3.1. Revenue Models of Supply Chain Members in the N Situation

In the *N* situation, suppliers and retailers in both supply chains do not invest in the BBTS. The supplier sells product i to its exclusive retailer at price $w_{i}^{N}$. The demand of the retailer is $D_{i}^{N}$, and the retail price of the product is $p_{i}^{N}$. Then, the expected returns of supply chain members $\pi_{ri}^{N}$ and $\pi_{mi}^{N}$ are as follows:(1)$$\pi_{mi}^{N}=(w_{i}^{N}-c_{m})D_{i}^{N}$$
(2)$$\pi_{ri}^{N}=(p_{i}^{N}-w_{i}^{N}-c_{r})D_{i}^{N}$$

Using backward induction to solve the retail price, we can get the retail price $p_{i}^{N}$ of agricultural products i. Then, putting the $p_{i}^{N}$ into Formula (1) provides the supplier’s equilibrium wholesale price $w_{i}^{N^{*}}$. According to the equilibrium wholesale price $w_{i}^{N^{*}}$, we can get the equilibrium retail price $p_{i}^{N^{*}}$. Putting the equilibrium retail price $p_{i}^{N^{*}}$ into the demand function $D_{i}^{N}$ provides the equilibrium demand $D_{i}^{N^{*}}$. Based on the equilibrium wholesale price $w_{i}^{N^{*}}$, the equilibrium retail price $p_{i}^{N^{*}}$ and the equilibrium demand $D_{i}^{N^{*}}$, we can derive the equilibrium incomes of the supplier and the retailer in Lemma 1.

**Lemma** **1.**
*When the two supply chains do not invest in the BBTS, the equilibrium demand is given by:*

(3)
$$D_{i}^{N^{*}}=-\frac{\left(\beta^{2}-2\right)\left(A+C_{1}+C_{2}+S_{1}^{N}+S_{2}^{N}\right)}{F}$$

*and the value function of the supply chain*

i

*is given by:*

(4)
$$\pi_{mi}^{N^{*}}=\frac{\left(\beta^{2}-2\right){\left(A+C_{1}+C_{2}+S_{1}^{N}+S_{2}^{N}\right)}^{2}}{BF}$$


(5)
$$\pi_{ri}^{N^{*}}=\frac{{\left(\beta^{2}-2\right)}^{2}{\left(A+C_{1}+C_{2}+S_{1}^{N}+S_{2}^{N}\right)}^{2}}{F^{2}}$$


*Furthermore, the profit of the whole AFSC is given by:*

(6)
$$\pi_{ri}^{N^{*}}+\pi_{mi}^{N^{*}}=\frac{2\left(\beta^{4}-5\beta^{2}+6\right){\left(A+C_{1}+C_{2}+S_{1}^{N}+S_{2}^{N}\right)}^{2}}{F^{2}}$$

*where*

(7)
$$A=-8a_{i}-6a_{j}\beta +3a_{i}\beta^{2}+2a_{j}\beta^{3}$$


(8)
$$C_{1}=+8c_{m}-9\beta^{2}c_{m}+2\beta^{4}c_{m}+8c_{r}-9\beta^{2}c_{r}+2\beta^{4}c_{r}$$


(9)
$$C_{2}=-2\beta c_{m}+\beta^{3}c_{m}-2\beta c_{r}+\beta^{3}c_{r}$$


(10)
$$S_{1}^{x}=-8\lambda_{i}^{x}q_{i}-6\beta \lambda_{j}^{x}q_{j}+3\beta^{2}\lambda_{i}^{x}q_{i}+2\beta^{3}\lambda_{j}^{x}q_{j}$$


(11)
$$S_{2}^{x}=+8kq_{j}+6\beta kq_{i}-3\beta^{2}kq_{j}-2\beta^{3}kq_{i}$$


(12)
$$B=4\beta^{4}-17\beta^{2}+16$$


(13)
$$F=4\beta^{6}-33\beta^{4}+84\beta^{2}-64$$



**Proposition** **1.**
*With an increase of*

$\lambda_{i}^{N}$

*in the profit of the supplier, the retailer and the whole AFSC experience increase. In addition, with the increase of*

κ

*, the profit of the whole AFSC reduces (see Appendix A for Proof of Proposition 1).*


According to Proposition 1, we find that when the two supply chains do not invest in the BBTS, the increase in the perceived quality safety factor raises the profits of AFSC members. In addition, the increase of the sensitivity coefficient of the market demand about the substitute quality reduces the profits of AFSC members.

### 3.2. Revenue Model of Supply Chain Members in the I Situation

In the I situation, due to the symmetry of two supply chains, we assume supply chain 1 invests in the BBTS, and supply chain 2 does not invest in the BBTS. By adopting the BBTS, the perceived quality safety factor of supply chain 1 is improved, and the supplier and retailer can effectively optimize the sales process and reduce sales cost (i.e., $\theta c_{m}$, $\theta c_{r}$) by the integration of enterprise information and the BBTS. The perceived quality safety factor of agricultural products in supply chain 1 is $\lambda_{1}^{I}$, while that of agricultural products in supply chain 2 is $\lambda_{2}^{I}$, and $\lambda_{1}^{I}>\lambda_{2}^{I}$. Revenues of supplier 1 and supplier 2 can be expressed as Formulas (14) and (15), respectively.
(14)$$\pi_{m1}^{I}=(w_{1}^{I}-\theta c_{m}-c_{or})D_{1}^{I}$$
(15)$$\pi_{m2}^{I}=(w_{2}^{I}-c_{m})D_{2}^{I}$$

Revenues of retailer 1 and retailer 2 can be expressed as Formulas (16) and (17), respectively.
(16)$$\pi_{r1}^{I}=(p_{1}^{I}-w_{1}^{I}-\theta c_{r}-c_{or})D_{1}^{I}$$
(17)$$\pi_{r2}^{I}=(p_{2}^{I}-w_{2}^{I}-c_{r})D_{2}^{I}$$

The demand $D_{i}^{I}$ in the I situation is given by:(18)$$D_{1}^{I}=a_{1}-p_{1}^{I}+\beta p_{2}^{I}+\lambda_{1}{}^{I}q_{1}-kq_{2}$$
(19)$$D_{2}^{I}=a_{2}-p_{2}^{I}+\beta p_{1}^{I}+\lambda_{2}{}^{I}q_{2}-kq_{1}$$

We solve the problem of agricultural products’ retail prices by backward induction, and get the retail prices $p_{1}^{I}$ and $p_{2}^{I}$ of agricultural products. Then, putting the $p_{1}^{I}$ and $p_{2}^{I}$ into (14) and (15) derives the suppliers’ equilibrium wholesale prices $w_{1}^{I^{*}}$ and $w_{2}^{I^{*}}$. Based on the equilibrium wholesale prices $w_{1}^{I^{*}}$ and $w_{2}^{I^{*}}$, we derive the equilibrium retail prices $p_{1}^{I^{*}}$ and $p_{2}^{I^{*}}$. Then, putting the equilibrium retail prices $p_{1}^{I^{*}}$ and $p_{2}^{I^{*}}$ into the demand functions (18) and (19) the equilibrium demand $D_{1}^{I^{*}}$ and $D_{2}^{I^{*}}$ are derived. Based on the equilibrium wholesale prices $w_{1}^{I^{*}}$ and $w_{2}^{I^{*}}$, the equilibrium retail prices $p_{1}^{I^{*}}$ and $p_{2}^{I^{*}}$, and the equilibrium demand $D_{1}^{I^{*}}$ and $D_{2}^{I^{*}}$, we can derive the equilibrium incomes of the supplier and the retailer in Lemma 2.

**Lemma** **2.**
*When supply chain 1 invests in the BBTS and supply chain 2 does not invest in the BBTS, the demand of supplier and retailer are given by, respectively:*

(20)
$$D_{1}^{I^{*}}=-\frac{\left(\beta^{2}-2\right)\left(A+\theta C_{1}+C_{2}+O_{1}+S_{1}^{I}+S_{2}^{I}\right)}{F}$$


(21)
$$D_{2}^{I^{*}}=-\frac{\left(\beta^{2}-2\right)\left(A+C_{1}+\theta C_{2}+O_{2}+S_{1}^{I}+S_{2}^{I}\right)}{F}$$


*The equilibrium revenues of suppliers are given by:*

(22)
$$\pi_{m1}^{I^{*}}=\frac{\left(\beta^{2}-2\right){\left(A+\theta C_{1}+C_{2}+O_{1}+S_{1}^{I}+S_{2}^{I}\right)}^{2}}{BF}$$


(23)
$$\pi_{m2}^{I^{*}}=\frac{\left(\beta^{2}-2\right){\left(A+C_{1}+\theta C_{2}+O_{2}+S_{1}^{I}+S_{2}^{I}\right)}^{2}}{BF}$$

*and the equilibrium revenues of retailers are given by:*

(24)
$$\pi_{r1}^{I^{*}}=\frac{{\left(\beta^{2}-2\right)}^{2}{\left(A+\theta C_{1}+C_{2}+O_{1}+S_{1}^{I}+S_{2}^{I}\right)}^{2}}{F^{2}}$$


(25)
$$\pi_{r2}^{I^{*}}=\frac{{\left(\beta^{2}-2\right)}^{2}{\left(A+C_{1}+\theta C_{2}+O_{2}+S_{1}^{I}+S_{2}^{I}\right)}^{2}}{F^{2}}$$


*Furthermore, the profit of the whole AFSC is given by:*

(26)
$$\pi_{r1}^{I^{*}}+\pi_{m1}^{I^{*}}=\frac{2\left(\beta^{4}-5\beta^{2}+6\right){\left(A+\theta C_{1}+C_{2}+O_{1}+S_{1}^{I}+S_{2}^{I}\right)}^{2}}{F^{2}}$$


(27)
$$\pi_{r2}^{I^{*}}+\pi_{m2}^{I^{*}}=\frac{2\left(\beta^{4}-5\beta^{2}+6\right){\left(A+C_{1}+\theta C_{2}+O_{2}+S_{1}^{I}+S_{2}^{I}\right)}^{2}}{F^{2}}$$

*where*

(28)
$$O_{1}=+8c_{om}-9\beta^{2}c_{om}+2\beta^{4}c_{om}+8c_{or}-9\beta^{2}c_{or}+2\beta^{4}c_{or}$$


(29)
$$O_{2}=-2\beta c_{om}+\beta^{3}c_{om}-2\beta c_{or}+\beta^{3}c_{or}$$


*Based on the equilibrium profits of AFSC members, we get Proposition 2.*


**Proposition** **2.**
*With the increase in perceived quality safety factor*

$\lambda_{1}^{I}$

*, the profits of supplier, retailer and the whole AFSC 1 increase (see Appendix A for Proof of Proposition 2).*


According to Proposition 2, we find that improving the perceived quality safety factor increases the profits of supplier, retailer and the AFSC members.

### 3.3. Revenue Model of Supply Chain Members in the M Situation

In the *M* situation, retailers and suppliers in both supply chains invest in the BBTS. The supplier sells product i to its exclusive retailer at price $w_{i}^{M}$. The demand is $D_{i}^{M}$, and the retail price of the product is $p_{i}^{M}$. Then, the expected returns of supply chain members $\pi_{ri}^{M}$ and $\pi_{mi}^{M}$ are as follows:(30)$$\pi_{mi}^{M}=(w_{i}^{M}-\theta c_{m}-c_{om})D_{i}^{M}$$
(31)$$\pi_{ri}^{M}=(p_{i}^{M}-w_{i}^{M}-\theta c_{r}-c_{or})D_{i}^{M}$$

Using backward induction to solve the retail price, we can get the retail price $p_{i}^{M}$ of agricultural products i. Then, putting the $p_{i}^{M}$ into (30) derives the supplier’s equilibrium wholesale price $w_{i}^{M^{*}}$. Based on the equilibrium wholesale price $w_{i}^{M^{*}}$, we can get the equilibrium retail price $p_{i}^{M^{*}}$ and then put the equilibrium retail price $p_{i}^{M^{*}}$ into the demand function $D_{i}^{M}$ to get the equilibrium demand $D_{i}^{M^{*}}$. Based on the equilibrium wholesale price $w_{i}^{M^{*}}$, the equilibrium retail price $p_{i}^{M^{*}}$ and the equilibrium demand $D_{i}^{M^{*}}$, we can derive the equilibrium incomes of the supplier and the retailer in Lemma 3.

**Lemma** **3.**
*When two supply chains invest in the BBTS, the demand is given by:*

(32)
$$D_{i}^{M^{*}}=-\frac{\left(\beta^{2}-2\right)\left(A+\theta \left(C_{1}+C_{2}\right)+O_{1}+O_{2}+S_{1}^{M}+S_{2}^{M}\right)}{F}$$

*and the incomes function of supply chain*

i

*is given by:*

(33)
$$\pi_{mi}^{M^{*}}=\frac{\left(\beta^{2}-2\right){\left(A+\theta \left(C_{1}+C_{2}\right)+O_{1}+O_{2}+S_{1}^{M}+S_{2}^{M}\right)}^{2}}{BF}$$


(34)
$$\pi_{ri}^{M^{*}}=\frac{{\left(\beta^{2}-2\right)}^{2}{\left(A+\theta \left(C_{1}+C_{2}\right)+O_{1}+O_{2}+S_{1}^{M}+S_{2}^{M}\right)}^{2}}{F^{2}}$$


*Furthermore, the profit of whole AFSC is given by:*

(35)
$$\pi_{ri}^{M^{*}}+\pi_{mi}^{M^{*}}=\frac{2(\beta^{4}-5\beta^{2}+6){\left(A+\theta \left(C_{1}+C_{2}\right)+O_{1}+O_{2}+S_{1}^{M}+S_{2}^{M}\right)}^{2}}{F^{2}}$$



**Proposition** **3.**
*With increase in the perceived quality safety factor*

$\lambda_{i}^{M}$

*, the profits of AFSC investing in the BBTS increase (see Appendix A for Proof of Proposition 3).*


### 3.4. Analysis of Equilibrium Investment Strategy of the BBTS

Based on the above hypothesis and analysis, we can easily find that the profits of each of the two AFSCs in the N situation are the same. In order to further study the equilibrium investment strategy of the BBTS, we analyzed the change of profits before and after investing in the BBTS by comparing the profits of supplier and retailer (i.e., situations I and M). We considered two conditions: (1) AFSC improve its profits by investing in the BBTS (i.e., *Cond*_1_: $\Delta \pi_{1}=\pi_{1}^{I}-\pi_{1}^{N}>0$; $\Delta \pi_{2}=\pi_{2}^{M}-\pi_{2}^{I}>0$); (2) AFSC hopes that they can make their profits higher than the other AFSC by investing in the BBTS (i.e., *Cond*_2_: $\pi_{1}^{I}>\pi_{2}^{I}$). (3) The profits of the two AFSCs after investing in the BBTS (situation M) are higher than before (situation N) (i.e., *Cond*_3_: $\pi_{i}^{M}>\pi_{i}^{N}$).

**Proposition** **4.**
*When $c_{om}+c_{or}<\frac{\lambda_{1}^{I}q_{1}-q_{2}\lambda_{2}^{I}}{\left(1+\beta \right)}+\left(c_{m}+c_{r}\right)\left(1-\theta \right)$, one of the two competitive AFSCs invests in the BBTS first, and the other AFSC invests in the BBTS, following its competitor, so the two competitive AFSCs invest in the BBTS (see Appendix A for Proof of Proposition 4).*


According to Proposition 4, we can obtain the threshold value of investment cost. The two competitive AFSCs invest in the BBTS when the sum of the investment costs is less than the threshold value. In addition, the threshold value sum of the BBTS costs $c_{om}+c_{or}$ has a negative correlation with the cost optimization coefficient θ and the sensitivity coefficient of the market demand about the substitute price β of the sales industry, based on the BBTS. This demonstrates that if investors or decision makers want to get a big threshold value sum of BBTS costs $c_{om}+c_{or}$, they should reduce the cost optimization coefficient θ and the sensitivity coefficient of the market demand about the substitute price β as much as possible.

Based on Proposition 4, we find that the perceived quality safety factor of supply chain 1 products $\lambda_{1}^{I}$ has a positive correlation with the threshold value sum of BBTS costs $c_{om}+c_{or}$, but the perceived quality safety factor of substitutes $\lambda_{2}^{I}$ has a negative correlation with the threshold value sum of the BBTS costs $c_{om}+c_{or}$. Namely, when the perceived quality safety factor, after investing in the BBTS, $\lambda_{1}^{I}$, is high, the AFSC has a wider range of acceptable investment costs $c_{om}+c_{or}$. If the perceived quality safety factor of substitutes without the BBTS is high, the investment cost should not be large.

In addition, if the sum of the BBTS costs $c_{om}+c_{or}$ meets condition *Cond*_3_ and does not meet condition *Cond*_2_, investing in the BBTS in the first place reduces profits, or makes the competitor more profitable than themselves. However, if the two competitive AFSCs both invest in the BBTS, their profits increase the same.

## 4. Discussion

In order to further study the changes in supply chain revenue under the three situations, we assumed a set of parameter values based on the previous literature in terms of marketing and operations management (P. Liu et al., 2020 [41]).

Based on the above analyses, we made $a_{1}=a_{2}=10$, $q_{1}=q_{2}=2$, κ=0.6. The perceived quality safety factor of the supply chain i before investing in the BBTS was $\lambda_{i}^{N}=0.7$. The perceived quality safety factor of the supply chain i improved, then we set $\lambda_{i}^{M}=0.8$ after investing in the BBTS. In addition, we assumed that the cost optimization coefficient was θ=0.75.

Figure 4, Figure 5 and Figure 6 show the changes of the retail prices, the wholesale prices and the demand with respect to some parameters. 

Figure 4 indicates that with the increase in the sensitivity coefficient of the market demand concerning the substitute price β, the retail price, the wholesale price and the demand increased in the proposed three situations (i.e., N, I and M). This might because the price competition expanded the market demand. Then, the retailer raises the retail price and the supplier also raises the wholesale price following the retailer.

Figure 5 indicates that with the increase in the perceived quality safety factor $\lambda_{i}^{M}$ after investing in the BBTS, the retail price, the wholesale price and the demand increased in the proposed three situations (i.e., N, I and M). It could be explained as follows: with the increase of the perceived quality safety factor $\lambda_{i}^{M}$ after investing in the BBTS, fresh agricultural products’ quality became more reliable. In general, consumers prefer products with reliable quality information. In addition, we could observe that, regarding supply chain 1 in the I situation and the two supply chains in the M situation, the retail price, the wholesale price and the demand were more sensitive to the changes of the perceived quality safety factor $\lambda_{i}^{M}$. This indicated that investing in the BBTS was beneficial to increase the optimal prices and expand demand.

Figure 6 indicates that with the increase in the sensitivity coefficient of the market demand for the substitute quality κ, the retail price, the wholesale price and the demand reduced in the proposed three situations (i.e., N, I and M). The reasons might be that the fierce quality competition increased the cost and reduced the market demand. Then, the AFSC members reduced its retail price and wholesale price.

Then, we used a numerical example to discuss the investment strategies of the BBTS. In general, we set the sensitivity coefficient of the market demand about the substitute price β is 0.5. Using the above parameter values, Figure 7a shows the changes of chain members profits after investing in the BBTS with the sum changes of the investment costs $c_{om}+c_{or}$. We can see that if AFSC members wanted to invest in the BBTS, the investment cost should be less than 0.3833. Here, the sensitivity coefficient of the market demand about the substitute price β was 0.5. In addition, Figure 7b shows the sum of the change trends of the BBTS investment costs $c_{om}+c_{or}$ following the change of the sensitivity coefficient of the market demand about the substitute price β. We found *Cond*_2_ < *Cond*_1_ < *Cond*_3_ as Proposition 4, and the minimum threshold value sum of the BBTS investment costs was 0.35.

Thus, we assumed $c_{om}=0.15$, $c_{or}=0.15$. Meanwhile, we assumed $c_{om}=0.22$, $c_{or}=0.22$ and $c_{om}=0.29$, $c_{or}=0.29$ to contrast with $c_{om}=0.15$, $c_{or}=0.15$.

When $c_{om}=0.15$, $c_{or}=0.15$, Figure 8 shows the changes in AFSCs’ profits with the sensitivity coefficient of the market demand for the substitute price β. We find that chain members’ incomes grew with the increase of the sensitivity coefficient of the market demand about the substitute price competition intensity. In addition, no matter what the value of β, the profits of AFSC increased after investing in the BBTS as Proposition 4 (i.e., $\pi_{ri}^{M}>\pi_{r1}^{I}>\pi_{r2}^{I}>\pi_{ri}^{N}$, $\pi_{mi}^{M}>\pi_{m1}^{I}>\pi_{m2}^{I}>\pi_{mi}^{N}$). The reason was that with the increase of β, the retail price, the wholesale price and the demand increased as observed in Figure 4.

When $c_{om}=0.22$ and $c_{or}=0.22$, the impacts of the sensitivity coefficient of the market demand about the substitute price on supply chain profits are shown in Figure 9. No matter what the value of β, the investment in the BBTS increased AFSCs’ profits. However, the profits of supply chain 2 would be higher than supply chain 1 when only supply chain 1 invested in the BBTS. Thus, when $c_{om}=0.22$ and $c_{or}=0.22$, AFSC would not invest in the BBTS as Proposition 4.

When $c_{om}=0.29$ and $c_{or}=0.29$, the increase in sensitivity coefficient of the market demand for the substitute price, β, results in the changes in AFSCs’ profits shown in Figure 10. Comparing Figure 9 and Figure 10, we found that when β was around 0.5, adding the BBTS investment costs would not always increase the benefits of supply chain members. When β was around 0.8, the benefits of chain members in Figure 10 were same as in the Figure 9. It could be explained by the fact that when β was around 0.8, $c_{om}=0.29$ and $c_{or}=0.29$ were less than $\frac{\left(8-3\beta^{2}\right)\left(\lambda_{i}^{I}-\lambda_{i}^{N}\right)q_{1}}{\left(8-9\beta^{2}+2\beta^{4}\right)}+\left(c_{m}+c_{r}\right)\left(1-\theta \right)$, but greater than $\min\left(\frac{\lambda_{i}^{I}q_{1}-q_{2}\lambda_{i}^{N}}{\left(1+\beta \right)}+\left(c_{m}+c_{r}\right)\left(1-\theta \right)\right)$.

We assumed the sum of investment cost $c_{om}+c_{or}$ was greater than its threshold value, but met condition *Cond*_3_. According to Figure 9 and Figure 10, we observe $\pi_{i}^{M}>\pi_{i}^{N}$ as Proposition 4. Although each of the two competitive AFSCs would not invest in the BBTS first, if the two AFSC could reach an agreement to invest in the BBTS together, it would be beneficial to their profits.

According to the above analyses, we proved Proposition 4 by numerical analysis. When $c_{om}+c_{or}$ met a certain range, the AFSC would invest in the BBTS, but, otherwise, they would not invest in the BBTS.

## 5. Conclusions and Future Researches

The BBTS plays a crucial role in the modern AFSC system. Compared with the traditional traceability service, which has some defects in the security of information storage, the BBTS can ensure the security of information storage. More and more enterprises are trying to invest in the BBTS. Walmart’s mango traceability system, based on blockchain technology, and the JD Zhizhen chain’s blockchain technology have proved that blockchain technology can improve the authenticity of product traceability information. Considering the investment cost of the BBTS, suppliers and retailers of agricultural products must make investment decisions in the face of industry competition.

In order to study the investment decision-making rules regarding the BBTS in competitive AFSCs, this paper designed two supply chains, each of which included a supplier and a retailer. Considering the new changes of consumers regarding perceived quality safety, the demand function of agricultural products was modified. Due to the symmetry of the two supply chains, the revenue model considered three proposed situations (N, l, M). The study found that:
(1)The increase of the sensitivity coefficient of the market demand for the substitute quality κ makes the retail price, the wholesale price and the demand reduce in the proposed three situations. If decision-makers can improve the perceived quality safety factor $\lambda_{1}^{I}$ and $\lambda_{i}^{M}$ by using the BBTS, it helps decision makers increase their benefits and the demand. The decision-makers of the BBTS should try their best to absorb the value of the BBTS in product quality traceability, and then improve the customers’ perceived quality safety factor after using the BBTS, and, finally, this helps decision makers increase their benefits.(2)The perceived quality safety factor after investing in the BBTS $\lambda_{1}^{I}$ has a positive correlation with the threshold value sum of the BBTS costs $c_{om}+c_{or}$, but the perceived quality safety factor without investing in the BBTS $\lambda_{2}^{I}$ has a negative correlation with the threshold value sum of the BBTS costs $c_{om}+c_{or}$. This demonstrates that when the decision makers want to invest in the BBTS, they should not only improve consumers’ perceived quality safety factor for their agricultural products $\lambda_{1}^{I}$, but also pay attention to consumers’ perceived quality safety factor for competitive products $\lambda_{2}^{I}$. When the competitor’s perceived quality safety factor $\lambda_{2}^{I}$ is high, investors should reduce investment costs $c_{om}+c_{or}$.(3)When the BBTS investment costs satisfy a certain condition, investing in the BBTS increases the profits of AFSC. In addition, the threshold value sum of the BBTS costs $c_{om}+c_{or}$ has a negative correlation with the cost optimization coefficient θ and the sensitivity coefficient of the market demand for the substitute price β. This demonstrates that if investors or decision-makers about the BBTS want the threshold value sum of the BBTS costs $c_{om}+c_{or}$ to be large, they should reduce the cost optimization coefficient θ and avoid fierce price competition as much as possible.(4)When the BBTS investment cost is greater than its threshold value, investments in the BBTS reduce chain members’ profits or make the competitor more profitable than themselves. Thus, no AFSC would invest in the BBTS when they cannot be sure whether their competitor would invest in the BBTS. If they can reach an agreement to invest in the BBTS together, the BBTS can still improve profits.

In this research, we arrived at some management suggestions. An AFSC should know that adopting the BBTS is mainly related to investment cost and the perceived quality safety level from customers. In addition, the AFSC should adopt price competition instead of quality competition.

This paper studied a simplified model of two competitive AFSCs, each of them was composed of a supplier and a retailer. An AFSC is a complex system. In future research, third-party logistics and multi-channel supply chains should be added to the model. In addition, the impression of uncertain demand on the income of the supply chain should be considered. Therefore, the next study will establish a multi-node model and consider uncertain requirements to study this problem.

## Figures and Tables

**Figure 1 foods-11-02981-f001:**
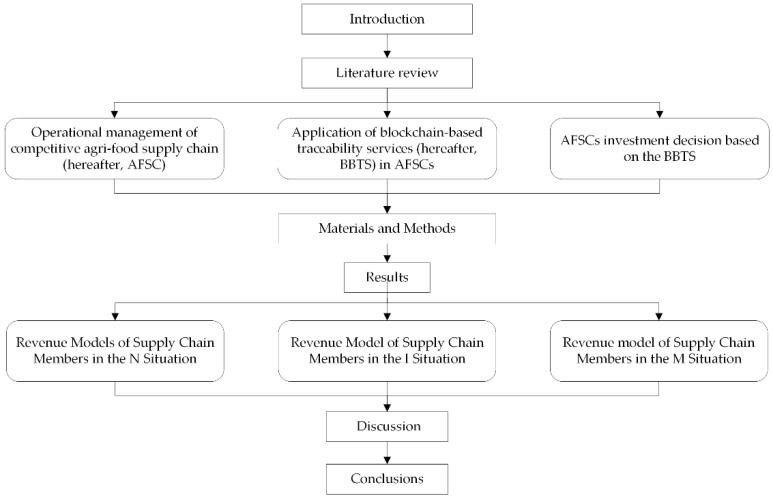
Article structure.

**Figure 2 foods-11-02981-f002:**
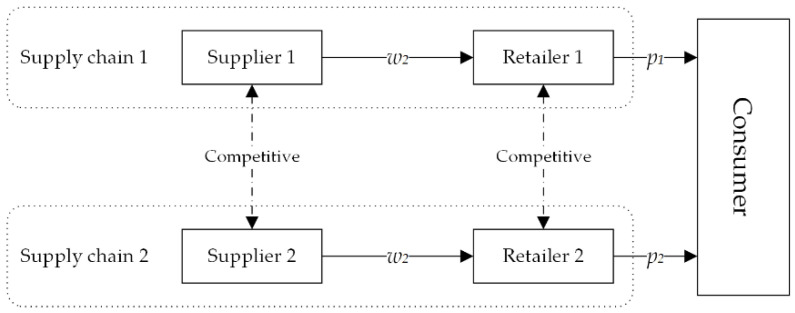
Competitive supply chain model.

**Figure 3 foods-11-02981-f003:**
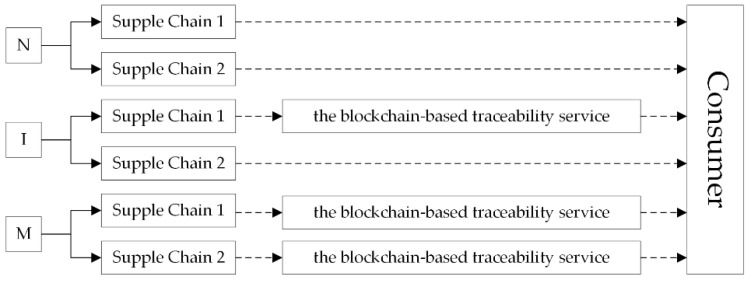
Three Investment situations of the BBTS.

**Figure 4 foods-11-02981-f004:**
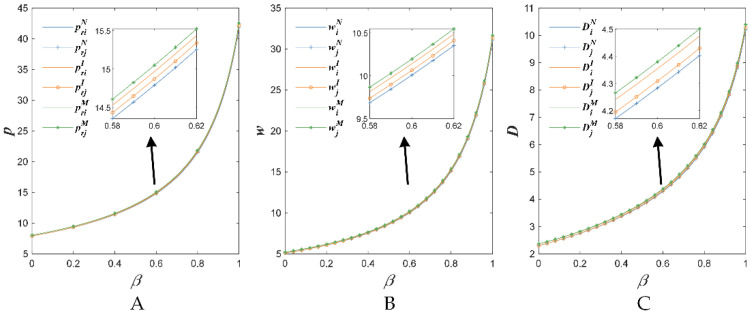
The sensitivity to the price of competitive agricultural products’ impact on retail price (**A**), wholesale price (**B**) and demand (**C**).

**Figure 5 foods-11-02981-f005:**
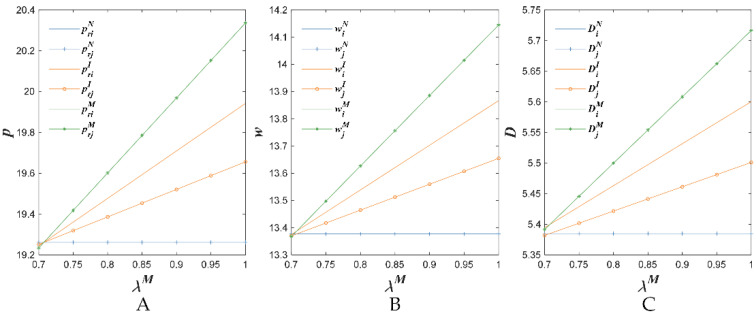
Quality safety perception coefficient’s impact on retail price (**A**), wholesale price (**B**) and demand (**C**).

**Figure 6 foods-11-02981-f006:**
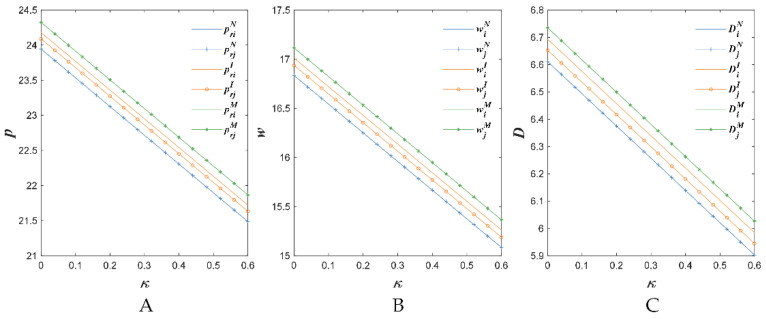
The sensitivity to the quality safety of competitive agricultural products’ impact on retail price (**A**), wholesale price (**B**) and demand (**C**).

**Figure 7 foods-11-02981-f007:**
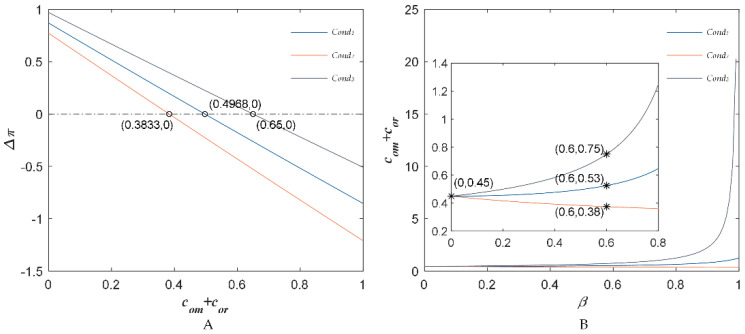
The relationship between total cost and profit (**A**), the sensitivity coefficient of competitive agricultural product price (**B**).

**Figure 8 foods-11-02981-f008:**
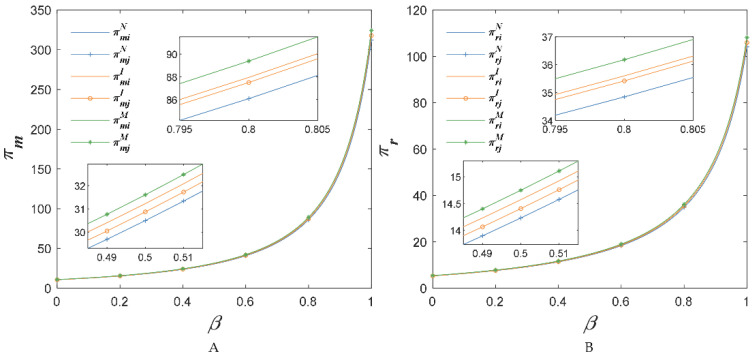
The price sensitivity coefficient of competitive agricultural products’ impact on income of suppliers (**A**) and retailers (**B**), when $c_{om}=0.15$, $c_{or}=0.15$.

**Figure 9 foods-11-02981-f009:**
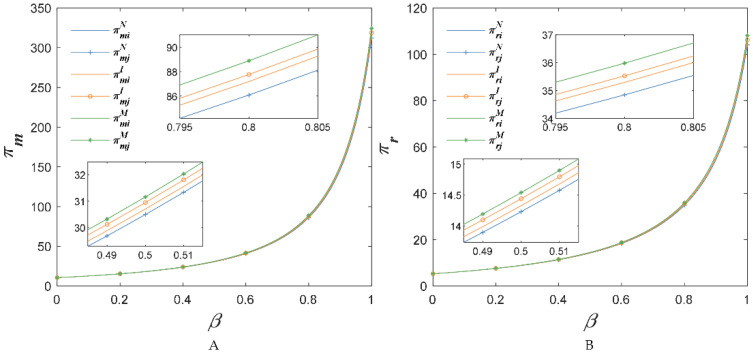
The price sensitivity coefficient of competitive agricultural products’ impact on income of suppliers (**A**) and retailers (**B**), when $c_{om}=0.22$ and $c_{or}=0.22$.

**Figure 10 foods-11-02981-f010:**
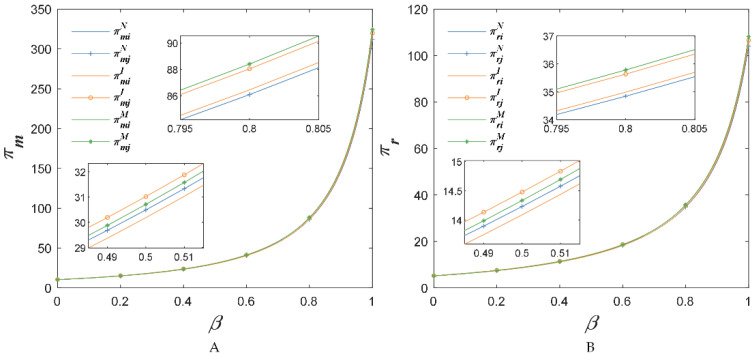
The price sensitivity coefficient of competitive agricultural products’ impact on income of suppliers (**A**) and retailers (**B**), when $c_{om}=0.29$ and $c_{or}=0.29$.

**Table 1 foods-11-02981-t001:** Variable description.

Variable	Explanation
x	Three different investment situations, Namely, x={N,I,M}.
i	Two kinds of substitutable agricultural products, i={1,2}.
j	Two kinds of substitutable agricultural products. Moreover, when i=1, j=2; when i=2, j=1.
$\pi_{ri}^{x}$	Revenues of the retailer in the x situation.
$\pi_{mi}^{x}$	Revenues of the supplier in the x situation.
$D_{i}^{x}$	The demand of i in the x situation.
$w_{i}^{x}$	The wholesale price of the i agricultural products in the x situation.
$p_{i}^{x}$	The retail price of the i agricultural products in the x situation.
$\lambda_{i}^{x}$	The perceived quality safety factor of the i agricultural products in the x situation.
$q_{i}$	The quality and safety level of agricultural products. This paper assumes that the quality and safety level of two kinds of agricultural products are equal $(q_{i}=q_{j}$).
$a_{i}$	The potential demand for the i agricultural products in the market.
β	The sensitivity coefficient of the market demand about the substitute’s price, the greater the β, the higher the sensitivity, and the stronger the competitiveness of the supply chain.
κ	The sensitivity coefficient of the market demand about the substitute’s quality.
$c_{r}$	The retailer’s cost of sales. This paper presumed that the costs of sales for the two products are the same.
$c_{m}$	The supplier’s cost of sales. This paper presumed that the costs of sales for the two products are the same.
$c_{or}$	The retailers’ the BBTS investment cost.
$c_{om}$	The suppliers’ the BBTS investment cost.
θ	The cost optimization coefficient. Through the integration of information and the BBTS from within the enterprise, suppliers and sellers can effectively optimize the sales process and reduce the sales cost. Therefore, we assume that the cost optimization coefficient is θ.
∗	Equilibrium solution

## Data Availability

Not applicable.

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
