# Peer review of "Investment Decision of Blockchain-Based Traceability Service Input for a Competitive Agri-Food Supply Chain"

_foods, 2022, doi:10.3390/foods11192981_

Round 1

Reviewer 1 Report

this is a novel research on the demand modelling for blockchain traceability services in the food supply chain. authors should pay attention to the figures, which are difficult to read and most of them do not have legends. Policy implications could be improved. If possible some actual data to test the model should be provided.

Author Response

Thanks very much for taking your time to review this manuscript. I really appreciate all your comments and suggestions! Please find my itemized responses in below and my corrections in the re-submitted files.

Reviewer 1

This is a novel research on the demand modelling for blockchain traceability services in the food supply chain. authors should pay attention to the figures, which are difficult to read and most of them do not have legends. Policy implications could be improved. If possible some actual data to test the model should be provided.

Response:We thank the reviewer for pointing this out. I am sorry that the figures in this manuscript are difficult to read. I have added captions to each figure for ease of understanding. At the same time, I added a table on page 7 to explain the variables to help read the figures.

Reviewer 2 Report

This is a paper modeling the value of investments on Blockchain traceability technologies within a supply chain. I find the approach interesting and innovative, and I also find the formality of the paper as a strength. Main areas of improvement to the paper are in the definition of the purpose and expected results, as well as in the explanation of models within the paper.

1. I find the abstract of the paper disconnected to the rest of the paper, or at least, I did not get a summary of the argument in the paper from the abstract. It needs to be revised so that it includes the main problem, purpose and results.

2. I find the research questions in the introduction much easy to understand, and I believe those are connected to models in the paper. I suggest to use them as a guide for the re-draft of the abstract.

3. Materials and methods need to be revised for clarity. For example, it is not clear the meaning of w and p in Figure 2, and although the authors explain that the figure shows how quality perception is improved by investing in BBTS, I an not sure I understand what elements of the figure describe this situation.

4. Not sure what are the hypotheses. I recommend to separate them from the text, make them as clear as possible... similar to the way you separate propositions as a separate paragraph. Not sure what is the meaning of all letters in the description of hypotheses (lamda, beta, a, p, q, etc.). Please describe their meaning in the text.

5. Numeric simulations in the discussion also need to be beter explained. I usually try to explain intuitively the meaning of the numbers. For example, authors suggest they picked number values from the literature, but what is the intuitive meaning of giving a1 and a2 a value of 10, and given that they did not explain the meaning of a, it is hard to assess the validity of any results.

6. Make sure you find an English language editor. The paper will benefit much from that work.

Author Response

This is a paper modeling the value of investments on Blockchain traceability technologies within a supply chain. I find the approach interesting and innovative, and I also find the formality of the paper as a strength. Main areas of improvement to the paper are in the definition of the purpose and expected results, as well as in the explanation of models within the paper.

  1. I find the abstract of the paper disconnected to the rest of the paper, or at least, I did not get a summary of the argument in the paper from the abstract. It needs to be revised so that it includes the main problem, purpose and results.

Response:Thank you for your comments on the abstract of this manuscript. Based on your comments, I have rearranged the main problem, purpose and results of the abstract.

Abstract: Competitive agri-food supply chain (AFSC) is an important component of AFSC. In a competitive environment, more and more AFSCs use blockchain traceability services to improve the traceability level of agricultural products. But what are the investment rules about BBTS and coordination rules in a AFSC are vital issues for many firms who want to adopt BBTS. To explore these laws, we constructed two competitive AFSCs, and each of them includes a supplier and a retailer. Considering the new changes in consumers' perception of product quality and safety after using the BBTS, the demand function was modified. Then we built the income functions of chain members under three situations of investment. The research finds that: 1) the improvement of customers’ perceived quality by using the BBTS can increase their benefits. 2) When decision-makers want to invest in the BBTS, they should pay attention to consumers’ perceived quality safety factor for their competitive products. 3) When the investment cost is greater than its threshold value, two competitive AFSCs should invest in the BBTS together.

  1. I find the research questions in the introduction much easy to understand, and I believe those are connected to models in the paper. I suggest to use them as a guide for the re-draft of the abstract.

Response:Thank you for your recognition of the introduction to this article. Based on your suggestion, I have revised the abstract with reference to the introduction.

  1. Materials and methods need to be revised for clarity. For example, it is not clear the meaning of w and p in Figure 2, and although the authors explain that the figure shows how quality perception is improved by investing in BBTS, I an not sure I understand what elements of the figure describe this situation.

Response:Thank you for pointing this out. In Figure 2, 'w' represents the wholesale price, and 'p' represents the retail price. I have added a table on the page 7 with a clear explanation of all the variables, hopefully it will help your reading. Figure 3 shows three situations which investing in BBTS improves quality perception.

  1. Not sure what are the hypotheses. I recommend to separate them from the text, make them as clear as possible... similar to the way you separate propositions as a separate paragraph. Not sure what is the meaning of all letters in the description of hypotheses (lamda, beta, a, p, q, etc.). Please describe their meaning in the text.

Response:I am sorry that the hypotheses of this manuscripts are not specified. The hypotheses represent the relationship between the variables of the AFSC, and the constraints on the demand model. I have extracted them according to the suggestion. In the table I added on the second page, I described in detail the meaning of the letters, such as 'lambda' represents the perceived quality safety factor, 'beta' represents the perceived quality safety factor, 'a' represents the potential demand for the agricultural products in the market, 'p' represents retail price, 'q' represents the quality and safety level of agricultural products.

  1. Numeric simulations in the discussion also need to be beter explained. I usually try to explain intuitively the meaning of the numbers. For example, authors suggest they picked number values from the literature, but what is the intuitive meaning of giving a1 and a2 a value of 10, and given that they did not explain the meaning of a, it is hard to assess the validity of any results.

Response:Thanks for your suggestion to the numerical simulation. Giving a1 and a2 a value of 10 represents the potential demand for the agricultural products in the market is 10. The meanings of other variable have been explained in new tables.

  1. Make sure you find an English language editor. The paper will benefit much from that work.

Response:Thank you for your valuable and thoughtful comments. We have carefully checked and improved the English writing in the revised.

Reviewer 3 Report

Dear All,

Although it is a well drafted manuscript, yet certain key literature is found missing:

1. https://doi.org/10.1177/0972150920907014

2. https://doi.org/10.4236/iim.2017.92004

3. https://link.springer.com/article/10.1007/s42488-021-00045-3

4. https://ieeexplore.ieee.org/abstract/document/9224613

Apart from that, the friction component hasn't been discussed (adaptability issues of new BBTS). Kindly incorporate these additions for the betterment of this manuscript.

Sincerely

Reviewer

Author Response

Although it is a well drafted manuscript, yet certain key literature is found missing:

Response:We have checked the literature carefully and added more references in the revised manuscript.

  1. https://doi.org/10.1177/0972150920907014

Response:This document has been cited on line 89, page 2.

  1. https://doi.org/10.4236/iim.2017.92004

Response:This document has been cited on line 85, page 2.

  1. https://link.springer.com/article/10.1007/s42488-021-00045-3

Response:This document has been cited on line 107, page 3.

  1. https://ieeexplore.ieee.org/abstract/document/9224613

Response:This document has been cited on line 118, page 3.

Apart from that, the friction component hasn't been discussed (adaptability issues of new BBTS). Kindly incorporate these additions for the betterment of this manuscript.

Response:We sincerely appreciate the valuable comments. I have discussed the adaptation of BBTS on line 153, page 4. 

We would like to express our great appreciation to you and reviewers for comments on our paper. Looking forward to hearing from you.

Thank you and best regards.

Round 2

Reviewer 2 Report

Thanks for the revisions of the paper